# Intestinal microbiology shapes population health impacts of diet and lifestyle risk exposures in Torres Strait Islander communities

Fredrick M Mobegi[1,2], Lex EX Leong[1], Fintan Thompson[1,3], Sean M Taylor[3], Linton R Harriss[3], Jocelyn M Choo[1,2], Steven L Taylor[1,2], Steve L Wesselingh[4], Robyn McDermott[3,5], Kerry L Ivey[1,6,7†]*, Geraint B Rogers[1,2†]*

[1]Microbiome and Host Health Programme, South Australian Health and Medical Research Institute, Adelaide, Australia; [2]SAHMRI Microbiome Research Laboratory, School of Medicine, College of Medicine and Public Health, Flinders University, Bedford Park, Australia; [3]Centre for Chronic Disease Prevention, Australian Institute of Tropical Health and Medicine, College of Public Health, Medical and Veterinary Sciences, James Cook University, Smithfield, Australia; [4]South Australian Health and Medical Research Institute, Adelaide, Australia; [5]School of Health Sciences, University of South Australia, Adelaide, Australia; [6]Department of Nutrition, Harvard T. H. Chan School of Public Health, Boston, United States; [7]Department of Nutrition and Dietetics, College of Nursing and Health Sciences, Flinders University, Adelaide, Australia

*For correspondence:
kivey@hsph.harvard.edu (KLI);
Geraint.Rogers@sahmri.com
(GBR)

†These authors contributed
equally to this work

Competing interests: The
authors declare that no
competing interests exist.

Reviewing editor: Jos WM van
der Meer, Radboud University
Medical Centre, Netherlands

**Abstract** Poor diet and lifestyle exposures are implicated in substantial global increases in non-communicable disease burden in low-income, remote, and Indigenous communities. This observational study investigated the contribution of the fecal microbiome to influence host physiology in two Indigenous communities in the Torres Strait Islands: Mer, a remote island where a traditional diet predominates, and Waiben a more accessible island with greater access to takeaway food and alcohol. Counterintuitively, disease markers were more pronounced in Mer residents. However, island-specific differences in disease risk were explained, in part, by microbiome traits. The absence of *Alistipes onderdonkii*, for example, significantly (p=0.014) moderated island-specific patterns of systolic blood pressure in multivariate-adjusted models. We also report mediatory relationships between traits of the fecal metagenome, disease markers, and risk exposures. Understanding how intestinal microbiome traits influence response to disease risk exposures is critical for the development of strategies that mitigate the growing burden of cardiometabolic disease in these communities.

## Introduction

Changes in dietary and lifestyle risk exposures are implicated in substantial global increases in the non-communicable disease burden (*World Health Organization, 2014*). This relationship has been particularly evident in remote and resource-limited communities, where increasing access to highly-processed, high-calorie foods, and alcohol (*World Health Organization, 2014*; *Popkin et al., 2012*) has been linked to a dramatic rise in metabolic syndrome and cardiovascular disease (*Yeates et al., 2015*; *Hu, 2011*). In Australia, for example, rates of chronic inflammatory and metabolic disease that are now three times higher in Aboriginal and Torres Strait Islanders populations than in their non-

Indigenous counterparts (*Zhao et al., 2008*; *Hodge et al., 2010*), with particularly high prevalence in very remote communities (*Arnold et al., 2016*; *Azzopardi et al., 2018*). This disparity in disease burden contributes substantially to health inequality (*Anderson et al., 2016*) and is replicated in remote Indigenous populations across the world (*Sonnenburg and Bäckhed, 2016*).

The role of host-microbiome interactions in shaping the impact of changing risk exposures is increasingly recognized (*Clemente et al., 2018*; *Wen and Duffy, 2017*; *Levy et al., 2017*). Intestinal microbial populations exert a profound influence over host metabolism, energy homeostasis, and innate and adaptive immunity (*Martin et al., 2019*; *Morgan et al., 2015*; *O'Keefe et al., 2015*), and are highly responsive to dietary and lifestyle exposures (*David et al., 2014*; *Engen et al., 2015*; *He et al., 2018*). Our understanding of these interactions is based on the analysis of predominantly urban, high income, European and North American populations. However, associations between the intestinal microbiome and population geography (*Brooks et al., 2018*), ethnicity (*Vangay et al., 2018*), and cultural background (*Queensland Health, 2017*) are well characterized. Since the influence of the microbiome on host physiology is likely to be similarly context-dependent, existing models of microbiome-mediation may not be appropriate for remote Indigenous communities in resource-limited settings. Instead, the complexity of interactions between diet, environment, lifestyle, human genetics, intestinal microbiology, and other factors, necessitate accurate modeling of changing patterns of disease at a community-level, and such efforts should arguably focus on those communities at greatest risk.

To investigate the role of the intestinal microbiome in defining the health impacts associated with changing risk exposures, we focused on two Indigenous communities in the Torres Strait Islands, an archipelago that lies between Australia's Cape York Peninsula and Papua New Guinea. Inhabitants of Mer (Murray Island), the most remote inhabited island in the Torres Strait, continue to consume a largely traditional diet, including local oily fish and garden produce. By contrast, those living in the regional center, Waiben (Thursday Island), have direct transport links to mainland Australia and ready access to takeaway food and alcohol. In keeping with the Indigenous people across Australia, the Melanesian Meriam people that populate these islands exhibit high rates of cardiovascular disease, type 2 diabetes, and cancer (*Australian Institute of Health and Welfare, 2018*). We hypothesized that intestinal microbiome characteristics would reflect assessed risk exposures, including age, diet, and lifestyle, and that a mediatory and modifying relationship between risk-associated microbiome traits and physiological markers of metabolic disease would be evident.

## Results

### Study population characteristics

According to Australian national averages (*Australian Bureau of Statistics, 2019*; *Shannon, 2002*), 9.1% of adult Australians consumed at least one sugar-sweetened beverage each day and 51% consumed at least two servings of fruit. Furthermore, 67% of the Australian adult population are classified as being overweight or obese, and 4.1% have type 2 diabetes. In contrast to these national patterns, 53% of the study cohort consumed at least one sugar-sweetened beverage each day. While 70% consumed at least two servings of fruit each day, 76% of the study population were overweight or obese and 39% had type 2 diabetes.

### Island-specific differences in exposure to chronic disease risk factors

Study participants were drawn equally from Waiben, the main commercial and administrative center of the Torres Strait Islands (n = 50), and Mer, a smaller, more remote, island (n = 50). There were no island differences in participant gender (males: n = 27 in Waiben, n = 24 in Mer, p=0.548) or in the prevalence of current smokers (smokers: n = 17 in Waiben, n = 16 in Mer, p=0.832). Residents of Mer largely consumed a diet that is traditional in the Torres Strait Islands (*Niccoli and Partridge, 2012*), including frequent consumption of fresh seafood. In comparison, Waiben residents had a more 'westernized' diet, characterized by significantly higher consumption of takeaway food and alcohol, and lower consumption of seafood (*Table 1*). Although current dogma suggests such dietary habits would be associated with increased chronic disease risk, increased levels of chronic disease risk markers were not observed for Waiben residents. Specifically, Mer residents had significantly (p<0.05) greater systolic blood pressure, mean arterial pressure (MAP), C-reactive protein, IL-12, IL-

**Table 1.** Characteristics of Waiben and Mer residents.

| Characteristic | Waiben (n = 50) | Mer (n = 50) | Unadjusted P-value * | Age-adjusted P-value † |
|---|---|---|---|---|
| *Age [years]* | 40 (31, 54) | 58 (36, 65) | **0.003** | NA |
| *Exposure chronic disease risk factors* | | | | |
| Sugar-sweetened beverage consumption [serves/day] | 1 (0, 1) | 1 (0, 1) | 0.576 | 0.562 |
| Fruit consumption [serves/day] | 2 (1, 3) | 2 (1, 2) | 0.680 | 0.682 |
| Vegetable consumption [serves/day] | 3.5 (2.0, 5.0) | 3.5 (2.0, 4.0) | 0.609 | 0.606 |
| Takeaway consumption [serves/week] | 1 (0, 2) | 0 (0, 1) | **0.011** | **0.009** |
| Seafood consumption [serves/week] | 2 (1, 3) | 2 (1, 5) | **0.020** | **0.021** |
| Alcoholic beverage consumption [serves/day] | 0.33 (0.00, 0.99) | 0.20 (0.00, 1.88) | 0.597 | 0.592 |
| Smoking [cigarettes/day] | 0.00 (0.00, 5.25) | 0.00 (0.00, 2.25) | 0.804 | 0.800 |
| Physical activity [min/week] | 180 (90, 236) | 210 (150, 360) | 0.925 | 0.925 |
| *Biological chronic disease risk factors* | | | | |
| Glucose [mmol/L] ‡ | 5.2 (4.7, 5.7) | 5.6 (4.8, 6.5) | 0.056 | **0.050** |
| HbA1c [mmol/mol] ‡ | 38 (34, 42) | 43 (36.5, 45) | 0.127 | 0.109 |
| Systolic blood pressure [mmHg] | 121 (112.5, 129.5) | 129 (115.3, 139.5) | **0.028** | **0.025** |
| Mean arterial pressure [mmHg] | 87.8 (72.0–113.7) | 95.8 (61.0–126.3) | **0.034** | **0.033** |
| Body mass index [kg/m$^2$] | 31.1 (25.8, 37.9) | 32.1 (25.5, 36.5) | 0.893 | 0.893 |
| Waist circumference [cm] | 105 (92, 116) | 116 (101, 125) | 0.019 | **0.016** |
| Waist to hip ratio | 0.62 (0.53, 0.69) | 0.68 (0.61, 0.75) | 0.035 | **0.027** |
| *Inflammatory biomarkers ‡* | | | | |
| Interferon γ [pg/mL] | 0.69 (0.61, 0.83) | 0.96 (0.67, 1.18) | **<0.001** | **<0.001** |
| C-Reactive protein [mg/L] | 0.36 (0.23, 0.51) | 1.36 (0.63, 1.76) | **<0.001** | **<0.001** |
| Interleukin-1β [pg/mL] | 0.39 (0.19, 0.61) | 0.37 (0.23, 0.5) | 0.534 | 0.536 |
| Interleukin-12p40 [pg/mL] | 1.98 (1.26, 3.47) | 6.32 (3.31, 11.37) | **<0.001** | **<0.001** |
| Interleukin-12p70 [pg/mL] | 0.46 (0.41, 0.51) | 0.82 (0.62, 1.07) | **<0.001** | **<0.001** |
| Interleukin-13 [pg/mL] | 1.72 (1.48, 1.94) | 1.82 (1.82, 2.07) | **0.008** | **0.008** |
| Interleukin-15 [pg/mL] | 7.37 (4.36, 10.7) | 8.76 (6.89, 12.23) | 0.281 | 0.272 |
| Interleukin-18 [pg/mL] | 11.34 (4.57, 18.17) | 21.38 (13.26, 37.51) | **<0.001** | **<0.001** |
| Interleukin-4 [pg/mL] | 4.97 (4.61, 6.09) | 6.80 (5.95, 9.23) | **<0.001** | **<0.001** |
| MCP-1/CCL2 [pg/mL] | 52.77 (32.89, 88.41) | 111.60 (67.65, 149.9) | **<0.001** | **<0.001** |
| Tumor necrosis factor α [pg/mL] | 0.17 (0.11, 0.30) | 0.47 (0.40, 0.68) | **<0.001** | **<0.001** |
| *Species-alpha diversity* | | | | |
| Count [Chao1 species richness] | 68.5 (64.0, 72.3) | 68.0 (57.8, 73.3) | 0.805 | 0.805 |
| Distribution [Shannon evenness index] | 0.68 (0.61, 0.71) | 0.71 (0.66, 0.75) | **0.027** | **0.026** |
| Variety [Shannon diversity index] | 2.83 (2.50, 2.98) | 2.94 (2.73, 3.20) | 0.061 | 0.060 |

Results are median (Q1, Q3).

* ANOVA between Mer and Waiben.

† ANCOVA between Mer and Waiben adjusted for age.

‡ analysis performed on log-transformed data. Abbreviations: MCP: Monocyte Chemoattractant Protein; MIP: Macrophage Inflammatory Protein.

18, and IFNγ, compared to Waiben residents. While not achieving statistical significance (p=0.100), the prevalence of T2D in Mer residents was also greater than in Waiben residents (48% versus 30%, respectively; *Table 1*).

The explanation for such a proatherogenic biomarker profile in Mer residents was not immediately apparent. Mer residents were significantly older than Waiben residents (mean age [years]: 41.5 versus 51.7; p=0.003), a recognized risk factor for chronic disease (*Arumugam et al., 2011*). However, significant differences in measures of blood pressure and inflammatory markers remained even after adjusting for age (*Table 1*).

## Fecal microbiome characteristics

Exploratory factor analysis of species relative abundance revealed six major factors (*Supplementary file 1a*). However, differences in microbiota traits were evident between the two island communities. Mer residents displayed higher species evenness (*Table 1*), even after adjustment for age (p=0.026). While the fecal microbiota composition for the study population as a whole was broadly in keeping with that described for adult populations more generally (*Human Microbiome Project Consortium, 2012*; *Gassasse et al., 2017*), the communities differed in their species-level microbiota composition (P[perm]=0.001, pseudo-F = 2.125, 9882 permutations); *Figure 1a*, *Figure 1—figure supplement 1*. Linear discriminant analysis effect size (LEfSe) identified differentially enriched species in each subpopulation (*Figure 1b*), which were phylogenetically diverse and represented five distinct bacterial phyla (*Figure 1c*). For example, the Proteobacteria and Euryarchaeota phyla were significantly higher in Mer compared to Waiben in both unadjusted and age-adjusted models (*Supplementary file 1b*). *Klebsiella pneumoniae* (NCBI: txid573; Proteobacteria) and *Escherichia coli* (NCBI:txid562; Proteobacteria), common causes of enteric and disseminated infection, and the important methanogen, *Methanobrevibacter smithii* (NCBI:txid2173; Euryachaeota), were also significantly enriched in Mer.

Characteristics of fecal microbiome metabolic capacity were then assessed. The prevalence of 18 microbial pathways, which accounted for 5% of the total community pathways identified by HUManN2, differed significantly between Mer and Waiben populations (*Figure 2a*). Three pathways that were more highly represented in the Mer population related to energy-generating glycolysis pathways, including homolactic fermentation (ANAEROFRUCAT-PWY: homolactic fermentation, p=0.005), glycolysis I (from glucose-6-phosphate, p=0.004), glycolysis II (from fructose-6-phosphate, PWY-5484, p=0.004), and glycolysis VI pathway (PWY66-400, p=0.004). Pathways that were more highly represented in the Waiben population included those relating to amino acid biosynthesis, including L-valine (VALSYN-PWY, p=0.006), L-isoleucine biosynthesis I and IV (ILEUSYN-PWY, p=0.006; PWY-5104, p=0.004), L-lysine biosynthesis VI (PWY-5097, p=0.009), and L-lysine, L-threonine and L-methionine biosynthesis II super pathway (PWY-724, p=0.004). In addition, pathways relating to aromatic acid biosynthesis (COMPLETE-ARO-PWY, p=0.004) and the precursor, chorismate (ARO-PWY, p=0.004 and PWY-6163, p=0.008) were also more prevalent in the Waiben population. Stratification of these pathways based on microbial taxa indicated that island-specific differences in glycolytic pathways were primarily accounted for by differences in the abundance of *E. coli*, while those relating to the biosynthesis of aromatic acid compounds were largely explained by the distribution of members of the *Roseburia* genus, including *Roseburia intestinalis* and *Roseburia hominis* (*Figure 2b*). The contribution of these bacterial taxa to the pathways identified was in keeping with the trends observed between the two island populations when metabolic capacity was assessed at a microbiota-wide level (FDR p<0.05; *Figure 2—figure supplement 1*).

## Microbiome as a mediator of exposure-inflammation relationships

From the data presented above, it is clear that our understanding of the factors that shape the microbiome composition and inflammatory profile of this population of Australian Torres Strait Islanders is incomplete. We, therefore, explored whether differences in the characteristics of the intestinal microbiome substantially influenced the relationship between exposure to dietary, lifestyle, and environmental factors (age, gender, island, smoking status and intakes of fruits, vegetables, takeaways, sugar-sweetened beverages, seafood, and alcohol) on circulating concentrations of inflammatory biomarkers. When comparing the various theoretical frameworks, structural equation modeling suggested that microbiome factors may potentially mediate the relation between

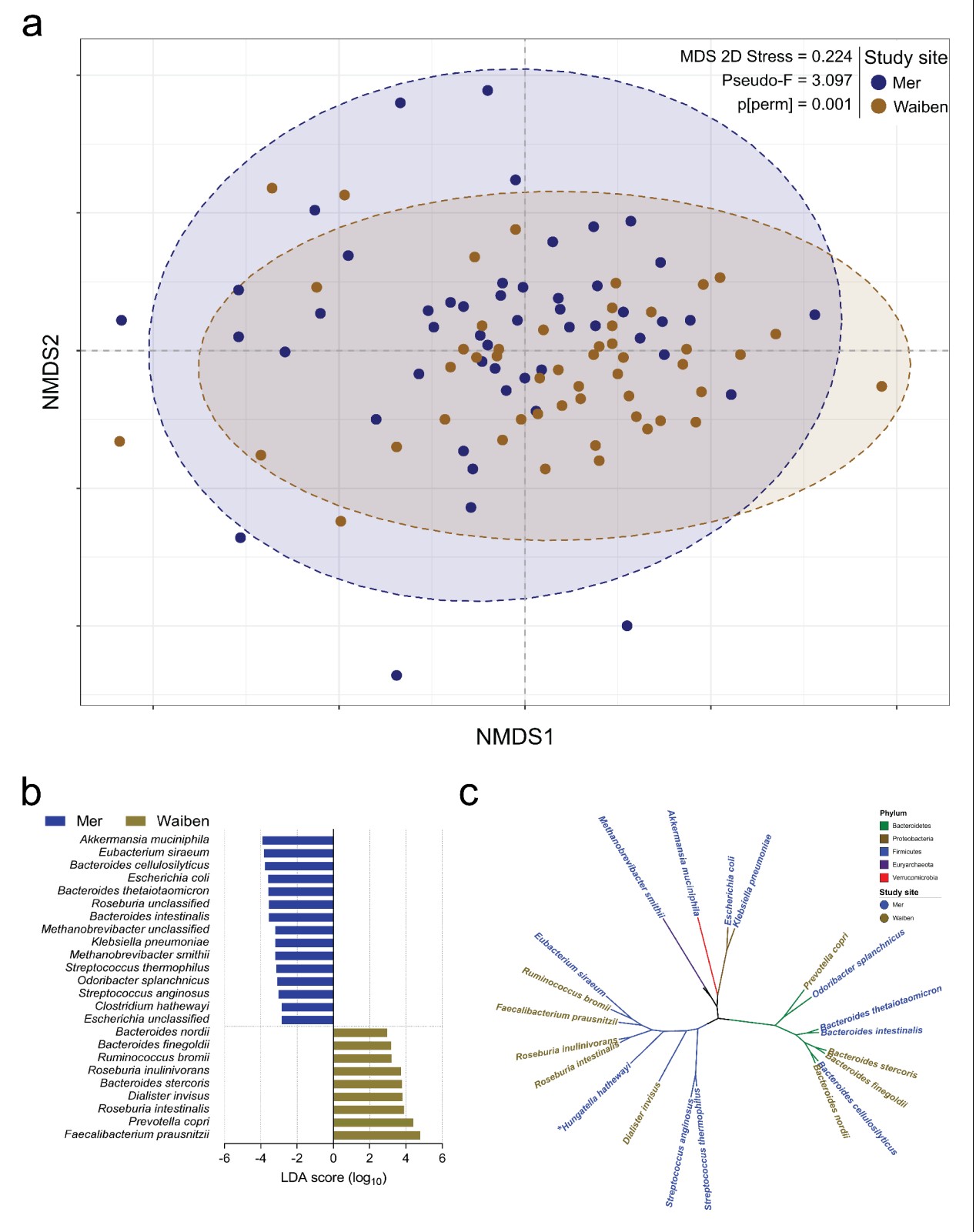

**Figure 1.** Island-specific differences in the species composition of the microbiome. (**a**) Non-metric Multidimensional Scaling of the Bray Curtis similarity resemblance matrix, with ellipses at 95% confidence interval. n = 100 (**b**) Fold differences in LDA mean proportions of differentially abundant species in Waiben and Mer. Bars were sorted based on the sequential ranking of significance and separated by the two populations. Only taxa significant at a logarithmic LDA score ≥3 and a factorial Kruskal-Wallis test Alpha (α)≤0.05 are shown. n = 100 (**c**) A phylogenetic tree based on the NCBI taxonomy of

*Figure 1 continued on next page*

*Figure 1 continued*

differentially abundant species identified using LEfSe. All species designated 'unclassified' were not used in generating the tree. Branches/edges are colored according to phylum rank classification and terminal nodes/species labels are colored based on study site overrepresentation. n = 100.

The online version of this article includes the following figure supplement(s) for figure 1:

**Figure supplement 1.** Taxonomic profile of the 100 Torres Strait Islander participants included in the study.

exposures and pathophysiology, when considering inflammatory profile (*Supplementary file 1c*, *Figure 3*, *Figure 3—figure supplement 1*). Most notable was the positive association sugar-sweetened beverage consumption had with the Biomarker Factor 3 (the latent construct representing IL13, IL15, IFN gamma, and IL1β), as well as with Species Factor 1, which consisted of *Haemophilus parainfluenzae*, *Bacteroides ovatus* and *fragilis*, members of the Veillonella genus, *Eubacterium halli*, *Anaerostipes hadrus*, *Roseburia intestinalis*, *Ruminococcus obeum*, *Ruminococcus gnavus*, Lachnospiraceae bacterium 5_1_63FAA and Lachnospiraceae bacterium 8_1_57FAA (*Figure 3*). Interestingly, a positive association was also observed between Biomarker Factor 3 and Species Factor 1. These associations were of particular interest, in part, because, when compared to the Australian national average, these communities consumed considerably more sugar-sweetened beverages. Of particular interest was the influence of Lachnospiraceae bacterium 8_1_57FAA (NCBI:txid665951) as in bootstrapped analyses of both islands combined, this was found to mediate the overall association between increased sugar-sweetened beverage consumption and increased serum concentration of IL-15 (*Figure 4a*, *Figure 4—figure supplement 1*). In fact, mediation by Lachnospiraceae bacterium 8_1_57FAA was found to significantly (p=0.044) account for (median [95% confidence intervals]) 13.5 (0.4, 44.2) percent of the total effect of sugar-sweetened beverages on IL-15 concentrations in unadjusted models, and account for 16.5 (0.6, 60.0) percent of the age-adjusted total effect (p=0.040). The mediation of the relationship between sugar-sweetened beverages and IL-15 remained significant, even after including island in the age-adjusted model, with Lachnospiraceae bacterium 8_1_57FAA significantly accounting for 21.2 (1.5, 78.9) percent of the total observed association

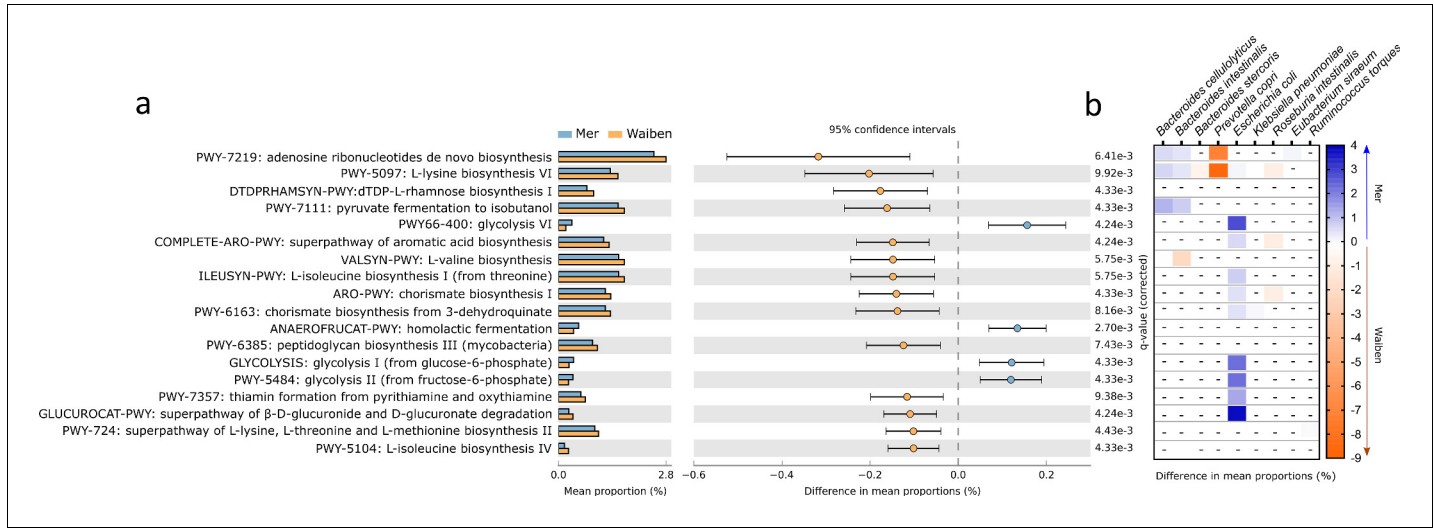

**Figure 2.** Functional pathways of the gut microbiota that significantly differed in abundance between Mer and Waiben communities. (**a**) Pathway abundances at the whole community level were statistically compared between the groups, and the 95% confidence interval of the effect-size for each pathway was determined using a non-parametric bootstrap method. (**b**) The heatmap represents community pathway abundances stratified into significant contributions from bacterial species to between-group differences. The color scale depicts the magnitude of the difference of stratified pathway abundances between the groups. All statistical comparisons were performed using the Mann-Whitney test and corrected for multiple testing using the false discovery rate method. Statistical significance for all comparisons was determined at p<0.05. Heatmap boxes with an '-' indicated non-significant stratified pathway abundance differences between the groups.

The online version of this article includes the following figure supplement(s) for figure 2:

**Figure supplement 1.** Heatmap representation of stratified contribution to whole-community pathway abundances.

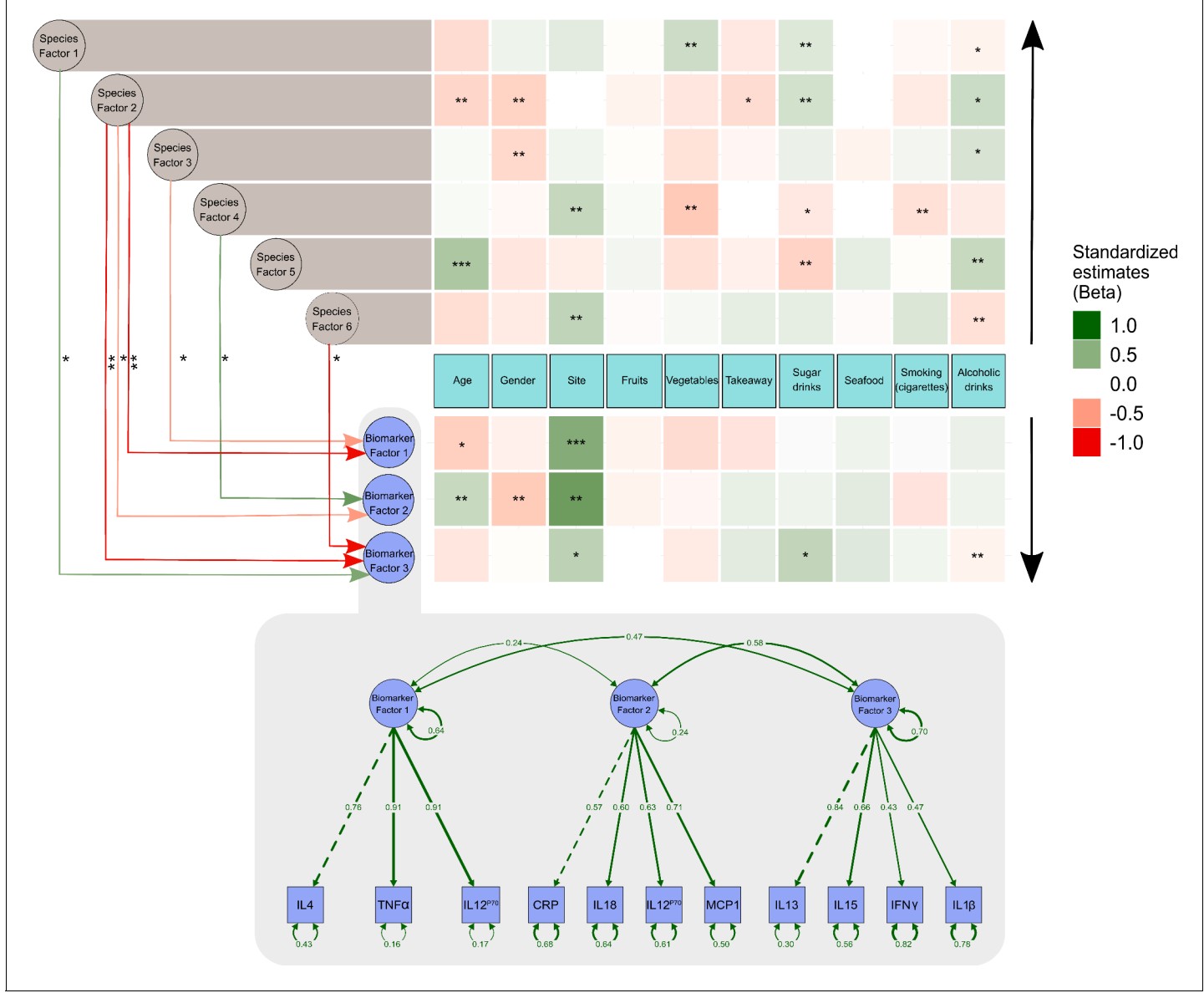

**Figure 3.** Structural equation model describing the pattern of interrelationships between exposures, microbiome, and host inflammation. A modified path diagram of the final SEM supporting the mediatory effect of the gut microbiota on inflammation. The heatmaps represent direct associations of diet, behavior, and demographics with the gut microbiota and inflammatory biomarkers. Only significant paths of the effect of the gut microbiota on inflammation are shown. Standardized β coefficients are reported. Species Factors 1–6 denote latent variables of the community gut microbiota modeled as exploratory factor analysis (EFA) regression scores of the species relative abundance; Biomarker Factors 1–3 denote latent variables representing community inflammatory biomarkers as grouped using EFA. *p 0.1–0.05, **p 0.05–0.001, ***p<0.001. Edge colors : Green: positive association; red: negative association. Black arrows indicate the direction of association of diet, demographics, and behavioral exposures with both inflammatory biomarkers and the gut microbiota.

The online version of this article includes the following figure supplement(s) for figure 3:

**Figure supplement 1.** Theoretical models for assessing the structure of associations between the various human exposures, the gut microbiota, and the host's inflammatory profile.

(p=0.032). Despite the total association being ameliorated, the moderation by Lachnospiraceae bacterium 8_1_57FAA remained significant even in the multivariate-adjusted model (p=0.036).

We then sought to identify the biological significance of this relationship by looking at the role of Lachnospiraceae bacterium 8_1_57FAA in mediating the observed relationship between sugar-sweetened beverage consumption and blood pressure (*Supplementary file 1d*). Mean arterial

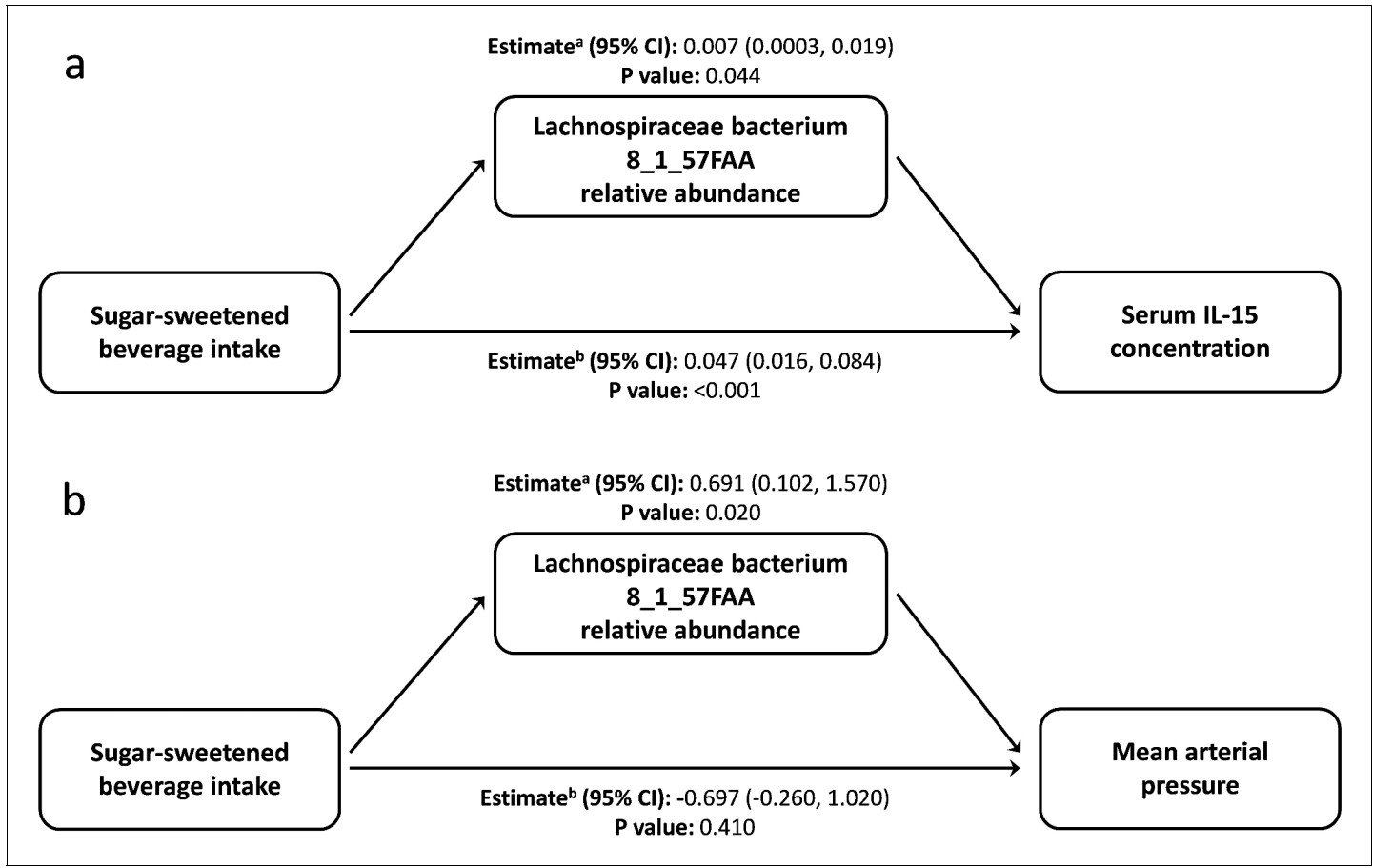

**Figure 4.** Lachnospiraceae bacterium 8_1_57FAA mediates the unadjusted relation of sugar-sweetened beverage intake with the serum concentration of Interleukin 15 and mean arterial pressure. (**a**) Lachnospiraceae bacterium 8_1_57FAA mediates the unadjusted relation of sugar-sweetened beverage intake with the serum concentration of Interleukin 15. The total effect (95% confidence interval) for the model was 0.054 (0.023, 0.090), P-value: <0.001. n = 100 (**b**) Lachnospiraceae bacterium 8_1_57FAA mediates the unadjusted relation of sugar-sweetened beverage intake with mean arterial pressure. The total effect (95% confidence interval) for the model was −0.006 (–1.530, 1.420), P-value: 0.950. n = 100 [a] ACME (average causal mediation effects); [b] ADE (average direct effects).

The online version of this article includes the following figure supplement(s) for figure 4:

**Figure supplement 1.** Role of Lachnospiraceae bacterium 8 1 57FAA in mediating the relation of sugar-sweetened beverage intake with the serum concentration of Interleukin 15.

**Figure supplement 2.** Role of *Lachnospiraceae* bacterium 8 1 57FAA in mediating the relation of sugar-sweetened beverage intake with the mean arterial pressure.

**Figure supplement 3.** 16S rRNA-based phylogeny of Lachnospiraceae.

pressure was greater in Mer than in Waiben in both unadjusted and age-adjusted models. Lachnospiraceae bacterium 8_1_57FAA mediated a positive association between sugar-sweetened beverage intake and MAP in unadjusted (*Figure 4b*) and age-adjusted models (*Figure 4—figure supplement 2*). Even after controlling for factors that could influence this relationship, Lachnospiraceae bacterium 8_1_57FAA continued to mediate a positive association between sugar-sweetened beverage intake and MAP for the combined island populations. In contrast to the significant mediatory effect of Lachnospiraceae bacterium 8_1_57FAA, no significant direct relationship between sugar-sweetened beverage intake and diastolic blood pressure was observed, indicating the considerable role played by the gut bacteria in the association of diet and pathophysiology in this context. The position of Lachnospiraceae bacterium 8_1_57FAA withing the Lachnospiraceae family is indicated in *Figure 4—figure supplement 3*.

## Microbiome as a moderator of exposure-pathophysiology relationships

Mediation is not the only way in which the microbiome can influence how dietary, lifestyle, and environmental factors influence pathophysiology. The microbiome can also interact with these relationships in such a way that the association of dietary, lifestyle, and environmental factors with pathophysiology depends on the presence/absence of a particular microorganism. Given that systolic blood pressure is an important chronic disease risk factor, and was higher in Mer than in Waiben systolic blood pressure was selected for an analysis of interaction. A microbiome-wide interaction study was performed, and, after correcting for multiple comparisons, *Alistipes onderdonkii* (NCBI:txid328813) was identified as a microorganism that significantly interacted with the island-systolic blood pressure relationship (*Figure 5*). The pattern of interaction was such that a failure to detect *Alistipes onderdonkii* in the gut microbiome was associated with increased systolic blood pressure, but only in Mer, and when these microorganisms were present, systolic blood pressure in Mer was identical to that of Waiben residents. These effect modifications remained even after adjusting for various diet and lifestyle exposure variables, suggesting that higher systolic blood pressure in Mer residents, despite Mer residents having lower traditional exposure risk factors, may be explained by an interaction between geographic location and the gut microbiome.

## Discussion

We report the intestinal microbiome to modulate cardiometabolic risk in two Torres Strait Islander populations. In particular, Lachnospiraceae 8_1_57FAA was found to mediate associations between sugar-sweetened beverage consumption and systemic inflammation and MAP in both communities assessed, while *Alistipes onderdonkii* was found to moderate systolic blood pressure, but only in residents of one of the two study islands.

Changes in diet and lifestyle are implicated in fundamental and ongoing shifts in population health (*World Health Organization, 2014*; *Samuel et al., 2008*). In particular, rates of cardiometabolic diseases are reaching epidemic proportions in many communities (*Yeates et al., 2015*; *Schutte et al., 2005*). Evidence has pointed to the gut microbiome as a critical determinant of response to risk exposures through its influence on host immune- and metabolic-regulation (*Thaiss et al., 2016*; *Kaibe et al., 2005*). Understanding the contribution of the gut microbiome to population health outcomes is particularly important in remote communities where chronic disease

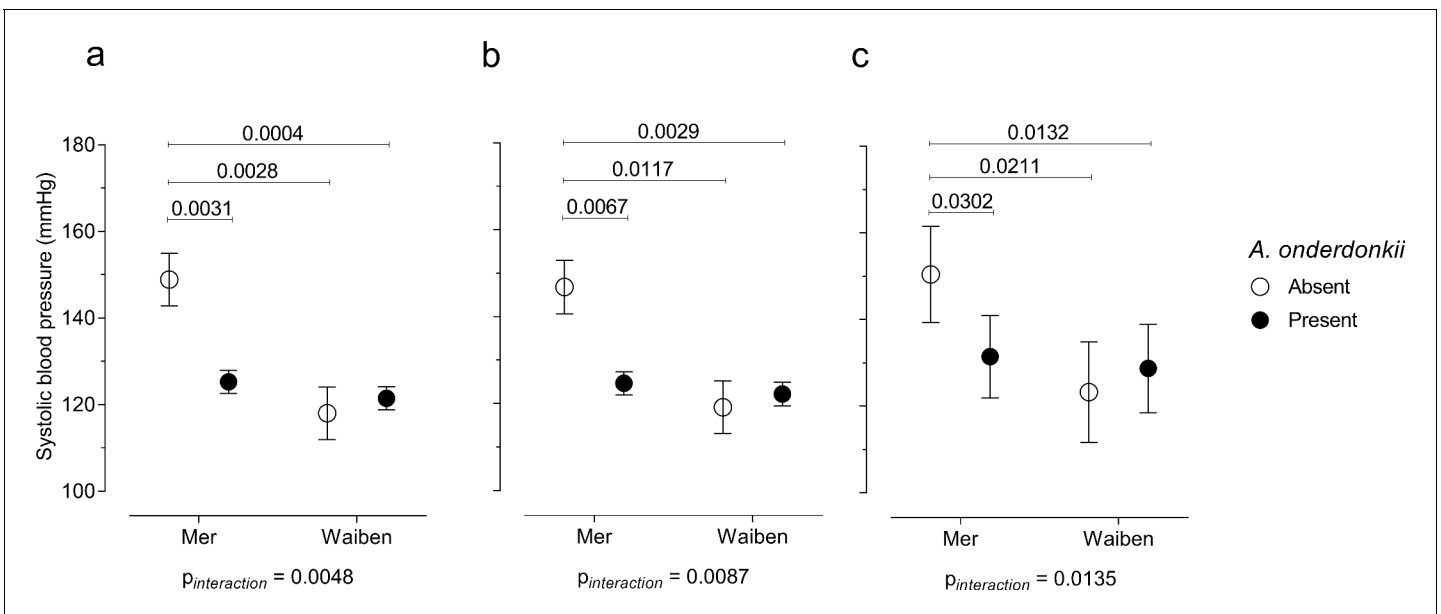

**Figure 5.** Systolic blood pressure in the context of an interaction between *Alistipes onderdonkii* and island systolic blood pressure. (a) Unadjusted model, (b) age-adjusted model, and (c) multivariate-adjusted model which includes the adjustment for age, body mass index, gender, smoking status, and intakes of fruits, vegetables, takeaways, sugar-sweetened beverages, seafood, and alcohol. Only significant P values (p<0.05) are shown n = 100.

burden is disproportionately high. However, models that have been developed using data from exclusively western populations may not be accurate in such contexts, given required assumptions about common risk exposures, disease susceptibilities, and intestinal microbiology. In our investigation of intestinal microbiology in two remote Indigenous communities, we identify fecal microbiome traits to be significant determinants of the association diet, lifestyle, and environmental factors have with disease risk.

The relationships between risk exposures, microbiome traits, and biological risk, that we report are highly biologically credible. For example, serum IL-15 concentration has been shown to be higher in patients with cardiovascular disease, compared to patients without the disease (*Krolopp et al., 2016*). IL-15 is a potent regulator of glucose uptake (*Kameyama and Itoh, 2014*), although whether its production is increased as a result of high sugar consumption is not currently well-understood. Lachnospiraceae bacterium 8_1_57FAA (NCBI:txid665951) was found to mediate the overall association between increased sugar-sweetened beverage consumption and increased serum concentration of IL-15, even after adjusting for multiple factors that may influence this mediatory relationship. We also observed a relationship between sugar-sweetened beverage consumption and diastolic blood pressure. Members of Lachnospiraceae, a family of bacteria in the order of Clostridiales and part of the normal human gut microbiota, have been implicated in the risk of metabolic (*Liu et al., 2018*; *Lippert et al., 2017*; *Schirmer et al., 2016*) and cardiovascular (*Lippert et al., 2017*) disease. In particular, associated Lachnospiraceae bacterium 8_1_57FAA has been associated with host cytokine levels (*Wang et al., 2019*) and BMI (*Turnbaugh et al., 2007*), although the underlying mechanisms that underpin these relationships remain poorly understood. Lachnospiraceae bacterium 8_1_57FAA is an uncultured taxon, identified through metagenomic analysis of human fecal material (*Jie et al., 2017*). It represents a little-known bacterium on the phylogenetic periphery of the Lachnospiraceae family. As applies to many of the taxa identified through metagenomics as influencing risk exposures-metabolic disease relationships, detailed characterization of this bacterium is required to better understand mediatory mechanisms.

Furthermore, when considering systolic blood pressure, we identified a significant interaction between *A. onderdonkii* and geographic location. When this taxon was present, no difference in systolic blood pressure between the residents of the two islands was observed. However, when *A. onderdonkii* was not detected in the gut microbiota of Mer residents, systolic blood pressure was significantly higher compared to residents of Waiben, suggesting that the absence of this bacteria was important for systolic blood pressure, but only within one island community. Members of the *Alistipes* genus are depleted in individuals with atherosclerotic cardiovascular disease (*Rowley et al., 2003*) and such an interaction could contribute substantially to the observed patterns of disease burden between the two islands that did not fit established models of risk.

While these findings contribute to our understanding of the fundamental relationships between commensal microbiology and non-communicable disease risk at a global level, it is important to be mindful of the fact that this study was undertaken because the cardiometabolic disease was identified as a matter of substantial and growing concern by the participating communities. It is therefore essential to consider how the insight gained from this analysis can be used to improve the health of the individuals living in these two communities. The existence of microbiome-modulation of non-communicable disease risk highlights the importance of access to food of high nutritional value, while identification of specific microbiome markers of increased disease susceptibility suggests a basis for more accurate prediction of an individual's relative disease risk. However, perhaps most exciting is our identification of moderation relationships, in which the absence of specific taxa, such as *A. onderdonkii*, increased disease risk within the exposure context of these communities.

The importance of understanding determinants of disease risk at a community level was further highlighted by the inflammatory profiles of the participants, as compared to those reported previously for remote Indigenous communities on the Australian mainland. For example, extremely high levels of CRP have been reported in remote Aboriginal communities (*Wang and Hoy, 2010*; *Miller et al., 2002*), with reported links to both dietary quality and risk of T2D (*Wang and Hoy, 2010*). By contrast, CRP concentrations amongst our study population were not markedly elevated, suggesting underlying differences in morbidity, potentially including lower rates of chronic infection.

Our understanding of how differences in microbiome traits influence host physiology, and ultimately, health outcomes, remains in its relative infancy. In part, this is due to the sheer complexity of microbiome-host interactions and the number of different potential points of direct or indirect

interaction between the gut microbes and the host. Analysis of the functional capacity of the microbiome (based on bacterial pathways encoded by the metagenome), enabled us to identify differences in functional capacity between study populations. However, functional redundancy within the gut microbiome, as well as widespread cross-metabolism, makes predicting the impact of such differences in overall microbiome output extremely difficult. We suggest that analysis of the gut metabolome could provide important insight into the nature of host-microbiome interactions. Furthermore, in addition to assessment of inflammatory cytokines, other factors that potentially contribute to diet-microbe-host interactions, such as plasma metabolites, warrant investigation.

A limitation of our study was its reliance on a relatively small sample, leading to an increased likelihood of type two statistical errors. However, we were still able to observe statistically significant associations that had biological relevance. Furthermore, it is important to note that the 100 participants in this study represented a substantial proportion of these two small island communities. Our study also employed a cross-sectional design, which meant that it was not possible to rule out the possibility of reverse causation and or establish causality. Given the contribution of genetic, developmental factors, and other unmeasured confounders to risk factor response, refinement of our models based on longitudinal analysis with a more complete set of exposure and outcome data would provide a valuable extension to this work.

Our findings highlight the potential contribution of the intestinal microbiome to community-specific patterns of cardiometabolic health through the mediation of relationships of specific diet, lifestyle, and environmental factors with host pathophysiology. These relationships have relevance for our understanding of changing burdens of non-communicable disease, particularly in remote, low-income, and Indigenous populations.

## Materials and methods

### Participants and setting

This observational study recruited participants from the Well Persons Health Check (WPHC). In 1997, the Australian government started a broad community health promotion and screening programme called the WPHC (*Berger et al., 2020*) conducted by the Torres and Cape Hospital and Health Service (TCHHS). Through WPHC several Indigenous (Aboriginal and/or Torres Strait Islander) communities in northern Queensland were screened annually for sexually transmitted infections and chronic cardiovascular as well as chronic metabolic conditions. In 2016, TCHHS collaborated with James Cook University (Zenadth Kes Health Partnership; *Weiland, 1978*) to conduct these surveys in two Australian island communities, Waiben and Mer. These islands are located in the Torres Strait Islands (TSI) archipelago, which lies in the waters separating the far northern continental Australia's Cape York Peninsula (northern-most tip of Queensland) and the Western Province of Papua New Guinea, off the continent of Australia. Mer (Murray Island) is a 4.3 km$^2$ remote island located on the eastern periphery of the Torres Strait region, 210 km northeast of Waiben and 340 km west of Port Moresby, with a population of 450 in 2016. The island is less connected with no known fast-food outlet. Waiben (Thursday Island) is a 3.5 km$^2$ landmass located about 39 km north of Cape York Peninsula with a population of 2938 in 2016. It is the administrative and commercial center of the Torres Strait Island Region.

In addition to the health check that was a standard component of the WPHC, participants consented to have additional health data collected for research purposes. Verbal consent was used whenever a written one could not be obtained. In addition to individual consent, parent/legal guardian consent was obtained for all participants younger than 18 years. The study incorporated all WPHC participants who identified as Aboriginal and/or Torres Strait Islander and attended the screening in October, November, and December 2016. There were 214 individuals aged 15 years and older who participated in the October, November, and December 2016 WPHC and identified as Aboriginal and/or Torres Strait Islander. Participants were excluded due to insufficient survey information (n = 2) and or not providing consent to participate in the research component of the health check (n = 1). Only participants who provided stool samples for metagenomic sequencing (total n = 100; 50 participants from each site) were then included in the current analysis. The study was granted ethical approval by the Far North Queensland Human Research Ethics Committee (HREC/

16/QCH/70–1059). In addition, written support from the local Community Council, Primary Health Care Service, and TCHHS was also provided (*Weiland, 1978*).

## Participant demographic, dietary, and behavioral factors

We collected the island of residence, age, gender, physical activity, and daily alcohol consumption as well as smoking status. Dietary intake of seafood, takeaway foods, vegetables, fruits, and sugar-sweetened beverages was determined using a culturally-appropriate food frequency questionnaire that assessed food and beverage consumption through the use of non-leading questions (*Weiland, 1978*).

Age, gender, height, weight, ethnicity, systolic blood pressure (SBP), diastolic blood pressure (DBP), known diabetes status, and known diabetes therapy were documented from WPHC clinical records. The mean arterial pressure (MAP) was calculated from brachial pressure values as follows:

$$MAP = \frac{SBP + (2 \times DBP)}{3}$$

Body mass index (BMI; kg/m$^2$) was computed from weight and height records. Hypertension was defined as SBP >140 and/or DBP > 90. Tobacco smoking and alcohol consumption were assessed using a standarized questionnaire.

## Clinical sample collection

A summary of key resources utilized, including sample collection kits, DNA isolation and library prep kits, DNA sequencing platform, and software is provided in *Supplementary file 1e* Each participant provided a sample of stool, blood, and urine. The stool was collected by participants using the OMNIgene GUT collection kit (DNA Genotek, Ontario, Canada), according to manufacturer's instructions. Briefly, the stool was deposited in a sterile container, transferred into the OMNIgene tube using the provided spatula, mixed with the stabilizing liquid, and stored at room temperature until collection. Blood and urine were collected by TCHHS as described previously (*Zhao et al., 2008*).

## Measurements of inflammatory and cardiometabolic markers

Concentrations of various biomarkers in blood and/or urine were measured using chemistry multiplex immunoassays and enzyme-linked immunosorbent assays (ELISA; *Bolger et al., 2014*) as follows: Interleukins (IL) IL-1β, IL-2, IL-4, IL-5, IL-6, IL-10, IL-12p40, IL-12p70, IL-13, IL-15, IL-17A, IL-18, and IL-33; type II interferon (IFNγ), monocyte chemoattractant protein 1 (MCP1/CCL2), macrophage inflammatory protein (MIP-1α/CCL3), and tumor necrosis factor alpha (TNFα) were measured using Luminex High Sensitivity (8-plex and 9-plex) Human ProcartaPlex Panels. Serum C-reactive protein (CRP) was measured using the CRP Human ProcartaPlex Simplex Kit. Serum lipopolysaccharide-binding protein (LBP) was also measured using Human LBP ELISA Kit. Random blood glucose (RBG) and glycated hemoglobin or hemoglobin A1c (HbA1c) were measured using blood glucose meter and Haemoglobin A1c (HbA1c) (Human) ELISA Kit, respectively. All commercial assays were performed according to manufacturers' instructions.

## DNA isolation and metagenomic sequencing

Microbial DNA was isolated from stool samples using DNeasy Powerlyzer PowerSoil Kit (Qiagen, Hilden, Germany). DNA yield was determined in Qubit Fluorometric Quantification using Quant-iT High-Sensitivity dsDNA Assay Kit. Libraries for shotgun metagenomic sequencing were fragmented using Nextera XT Library Preparation Kit v2 (Illumina Inc, San Diego, CA) and indexed using the Illumina Nextera XT Index Kit v2 according to manufacturer's instructions. Amplicon libraries were then sequenced on the Illumina HiSeq 2500 platform at the SAHMRI David R Gunn Genomics Suite using Illumina HiSeq SBS 2 × 125bp v4 kit (Illumina Inc). Reads were converted into FASTQ format using bcl2fastq v.1.8.4. On average, 16,666,136 sequencing reads were obtained from each sample before quality filtering.

## Bioinformatic processing

Paired-end reads were adapter- and quality-filtered using Trimmomatic v.0.36 (*Truong et al., 2015*) with the following parameters: PE ILLUMINACLIP:TruSeq3-PE:3:30:10 LEADING:10 MINLEN:30 SLIDINGWINDOW: 4:30; HEADCROP:10. Samples included in the downstream analysis had at least 7.4 million paired-end reads of at least 91 bp in length after quality control. The quality of reads was checked before and after this process using FastQC v.0.11.5.

Using the quality-controlled paired-end reads, taxonomic profiling was performed using the metagenomic phylogenetic analysis (MetaPhlAn) v.2.6 (*Franzosa et al., 2018*), which uses a database of clade-specific markers to quantify microbiota constituents at the species and higher taxonomic levels. The generation of the abundance of metabolic pathways was performed using HUMAnN2 (*Asnicar et al., 2015*). Taxonomic and metabolic profiles for both islands were visualized using GraPhlAn v.0.9.7 (*Whittaker, 1972*). Sequenced metagenomic data were submitted to the National Center for Biotechnology Information (NCBI) Sequence Read Archive (SRA) under BioProject ID PRJNA503909. Summary of the quality-controlled fecal microbiome sequencing reads is shown in *Supplementary file 1f*.

## Statistical analysis

Statistical analyses and graphical illustrations were accomplished in R software for statistical computing, Primer 7, GraphPad Prism 7, and Statistical Analysis Software (SAS/STAT) suite. Rare taxa, present in less than 40% of the participants at <1% relative abundance, were excluded from the analysis.

Alpha-diversity is defined as a local measure of average species diversity in a community. (*Whittaker, 1972*; *Jost, 2007*). To estimate the species alpha diversity in Mer and Waiben, we examined the microbial composition in each community using the *vegan* (*Oksanen et al., 2018*) package in R. (*R Development Core Team, 2010*) Accordingly, the '*estimate*' function was used to calculate Chao1's measure of richness; a measure of the number of species (or other taxonomic levels) present at a site. The '*diversity*' function was used to estimate the Shannon-Wiener Index (*H'*); a measure of how many different types of species are present in the community. Species diversity takes into account both species richness and the dominance or evenness of the species. Evenness measures homogeneity in a community in terms of species relative abundances. A community in which all species are proportionately common is regarded as stable and has a high degree of evenness. Since *vegan* does not provide a function for Shannon's evenness/equitability (*E*), we used a modified R function to estimated *E* by dividing the actual diversity (*H'*) with the maximum possible diversity value ($H_{max}$). The reasoning is, *H'* decreases when the species composition is more uneven and reaches a maximum with the highest evenness. Therefore, when all species are equally common, $H_{max} = \ln S$ (where *S* is the total number of species), and $E = H' \div H_{max}$. Differences in species composition within Mer and Waiben were analyzed in GraphPad using the Mann-Whitney *U*-test the with FDR multiple-testing adjustment.

Beta diversity is defined as the change in species composition from one community to the next (*Whittaker, 1972*; *Jost, 2007*). Species relative abundances were square root-transformed and used to calculate Bray-Curtis distances of samples dissimilarity in Primer6. To test the null hypothesis of no difference in species relative abundance between islands, we used permutational multivariate analysis of variance (PERMANOVA) model on the Bray-Curtis dissimilarity data under a partial (type III) sum of squares, a fixed-effects sum to zero for mixed terms, 10,000 unrestricted parameters permutation of residuals and Monte Carlo tests in Primer v.6. The matrix of Bray-Curtis distances was also used as input to calculate non-metric multidimensional scaling (NMDS) coordinates using *vegan* package in R (*R Development Core Team, 2010*). The first and second components were used to create an ordination plot using ggvegan.

Linear discriminant analysis (LDA) effect size (LEfSe) v.1.0 (*Segata et al., 2011*) was used to determine the species typifying the disparities between Mer and Waiben populations. To achieve this, a table of pre-filtered species relative abundances was normalised to ensure values for each species across all samples summed up to 1 million. Island of sample origin was used as a class variable. Nonparametric factorial Kruskal-Wallis (KW) sum-rank tests were then performed to detect species with significant ($\alpha < 0.05$) differential abundance with respect to the class. The species that were significantly different between islands were used to build an LDA model from which the relative difference among classes was used to rank the features. The threshold for the logarithmic LDA score was set

at ≥3. Finally, the ranked effect-size values were visualized in a LEfSe plot. Significant functional pathways were stratified into bacterial species, and the resulting stratified pathways were filtered to include only those that were present in at least 80% in either the Mer or Waiben populations. Stratified pathway abundances that remain significantly different between Mer and Waiben were determined using the Mann-Whitney $U$-test with the FDR multiple-testing adjustment, at a significance threshold of 0.05.

Correlations between measurements of each circulating biomarker in the host and the taxonomic or functional profiles of the microbiota were analyzed using Spearman's rank correlation implemented in the *Hmisc* package in R and visualized using *ggplot2* and *gplots* packages in R. Significance thresholds for the coefficients of association and *P-value*s were respectively set at rho ($\rho$) ≥ | 0.3| and p-value≤0.05, Benjamini-Hochberg (BH) adjusted for multiple hypothesis testing.

Structural Equation modeling (SEM) was performed in R using the *lavaan* package (*Rosseel, 2012*) to assess the direct and indirect associations of host diet, behavior and demographics with host biomarkers of inflammation. Initial path diagrams were visualized using *semPlot* (*Epskamp and Epskamp, 2017*) package in R and partly modified into heatmaps using *ggplot2*. Briefly, we modeled the inflammatory biomarkers using exploratory factor analysis to identify three latent (unobserved) constructs, with TNF-$\alpha$, IL-4, IL-12$^{P70}$, as indicators for the first latent variable (Biomarker Factor 1), CRP, IL-18, and MCP1 as indicators for the second latent variable (Biomarker Factor 2), and IL-15, IL-33, IFN$\gamma$, and IL-1$\beta$ indicators for the third latent variable (Biomarker Factor 3). Observed variables potentially important to inflammation, including alcohol consumption, smoking, diet (fruits, vegetables, fish, and intake of fatty and sugary foods typical of takeaways and soft-drinks), and demographics (age, gender, BMI, and island/study site) were also included. Four possible association frameworks were developed to explain the link between the specific exposures (age, gender, island, smoking status and intakes of fruits, vegetables, takeaways, sugar-sweetened beverages, seafood, and alcohol), the microbiome, and host physiology (*Figure 3—figure supplement 1*): Framework (1) the gut microbiome mediates the exposure-inflammation relationship; Framework (2) the gut microbiome influences pathophysiology but is not influenced by risk exposures; Framework (3) the gut microbiome is influenced by exposures but does not influence pathophysiology; and Framework (4) exposure risk factors are associated with the inflammatory profile, and the inflammatory profile predicts the gut microbiome composition. The model was assessed using the confirmatory factor analysis (CFA) function and estimated using the robust variant of maximum likelihood parameter estimation (MLR) with robust (Huber-White) standard errors and default bootstrapping (asymptomatically equivalent to Yuan-Bentler correction statistic) to account for data non-normality, presence of dichotomous and categorical variables, and correction for multiple hypothesis testing. Equivalents of the final SEM model were investigated and defined by adjusting the directionality of correlation or inferring co-variation instead of causation. Standard SEM fit guidelines (*Hooper and Coughlan, 2008*; *Hu and Bentler, 1999*), that is, chi-squared *P-value*; ($\chi^2$ *P*-value) >0.05; comparative fit index (CFI) >0.8; standardized root-mean residuals (SRMR) <0.08; root-mean-square error of approximation (RSMEA) <0.05; and the lowest Akaike information criterion (AIC) were used to assess model-data fit for each of the four theoretical frameworks.

Individual mediation relationships between host exposures, host gut microbiology (as the mediator), and host pathophysiology (as the outcome) were evaluated using the 'mediation' package in R. Fitted objects of the exposure and the mediator were analyzed using the 'mediate' function with nonparametric bootstrapping methods. Interaction was assessed through the incorporation of an interaction term into regression models.

## Acknowledgements

This work was supported by the National Health and Medical Research Council [grant number GNT0631947]. The salary of KLI was supported by a National Health and Medical Research Council fellowship and GBR was supported by a Matthew Flinders Research Fellowship and a National Health and Medical Research Council Senior Research Fellowship. The Zenadth Kes Health Partnership Project was made possible by the contribution of many people, including the diligent team from Torres and Cape Hospital and Health Service, James Cook University staff and students, Waiben and Mer community members, and the many people that generously gave their time to participate in the study.

## Additional information

### Funding

| Funder | Grant reference number | Author |
|---|---|---|
| National Health and Medical Research Council | GNT0631947 | Robyn McDermott |
| National Health and Medical Research Council | | Geraint Rogers |

The funders had no role in study design, data collection and interpretation, or the decision to submit the work for publication.

### Author contributions

Fredrick M Mobegi, Data curation, Formal analysis, Writing - original draft; Lex EX Leong, Data curation, Formal analysis, Writing - review and editing; Fintan Thompson, Sean M Taylor, Linton R Harriss, Data curation, Project administration, Writing - review and editing; Jocelyn M Choo, Formal analysis, Writing - review and editing; Steven L Taylor, Steve L Wesselingh, Writing - review and editing; Robyn McDermott, Project administration, Writing - review and editing; Kerry L Ivey, Conceptualization, Writing - original draft, Writing - review and editing; Geraint B Rogers, Conceptualization, Writing - original draft, Project administration, Writing - review and editing

### Author ORCIDs

Fredrick M Mobegi (iD) https://orcid.org/0000-0003-0554-9919
Steven L Taylor (iD) https://orcid.org/0000-0003-4357-8243
Kerry L Ivey (iD) https://orcid.org/0000-0001-9855-2599

### Ethics

Human subjects: In addition to the health check that was a standard component of the WPHC, participants consented to have additional health data collected for research purposes. Verbal consent was used whenever a written one could not be obtained. In addition to individual consent, parent/legal guardian consent was obtained for all participants younger than 18 years. The study was granted ethical approval by the Far North Queensland Human Research Ethics Committee (HREC/16/QCH/70-1059). In addition, written support from the local Community Council, Primary Health Care Service and TCHHS was also provided.

### Decision letter and Author response

Decision letter https://doi.org/10.7554/eLife.58407.sa1
Author response https://doi.org/10.7554/eLife.58407.sa2

## Additional files

### Supplementary files

• Supplementary file 1. Intestinal microbiology shapes population health impacts of diet and lifestyle risk exposures in Torres Strait Islander communities: Supplementary files.

• Transparent reporting form

• Reporting standard 1. STROBE checklist.

### Data availability

Shotgun metagenomic sequence data reported in this study were submitted to the National Center for Biotechnology Information (NCBI) Sequence Read Archive (SRA) archive and are publicly available via the accession numbers SRA: SRP167939 and BioProject: PRJNA503909.

The following dataset was generated:

| Author(s) | Year | Dataset title | Dataset URL | Database and Identifier |
|---|---|---|---|---|
| Mobegi F | 2018 | Torres Strait Islands Health Surveys, 2016 | https://www.ncbi.nlm.nih.gov/bioproject/PRJNA503909 | NCBI BioProject, PRJNA503909 |

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
