## [Decision Letter]

**Acceptance summary:**

This is an important study in the Australian context where indigenous inhabitants have a markedly reduced life span and an excess of multiple diseases. The bacterial composition of the intestine, the microbiome, is influenced by diet, environment and lifestyle. The microbiome is important for health and disease. The authors investigated the microbiome in two Torres Strait island populations with distinctly different food access, and lifestyle. They found significant differences in the microbiome between the communities. The study provides insight into interactions between the microbiome, high sugar beverage consumption and inflammation. Also between the microbiome, these beverages and high blood pressure, associations were found. Interestingly the paper goes against the current belief that western diet has worse effects on metabolism than traditional diet. The authors argue that the models that have been developed using data from exclusively western populations may not be accurate in remote communities where chronic disease burden is disproportionately high.

**Decision letter after peer review:**

Thank you for submitting your article "Intestinal microbiology shapes population health impacts of diet and lifestyle risk exposures in remote communities" for consideration by *eLife*. Your article has been reviewed by three peer reviewers, and the evaluation has been overseen by Jos van der Meer as the Senior and Reviewing Editor. The reviewers have opted to remain anonymous.

The reviewers have discussed the reviews with one another and the Reviewing Editor has drafted this decision to help you prepare a revised submission.

As the editors have judged that your manuscript is of interest, but as described below that additional experiments or analyses are required before it is published, we would like to draw your attention to changes in our revision policy that we have made in response to COVID-19 (https://elifesciences.org/articles/57162). First, because many researchers have temporarily lost access to the labs, we will give authors as much time as they need to submit revised manuscripts. We are also offering, if you choose, to post the manuscript to bioRxiv (if it is not already there) along with this decision letter and a formal designation that the manuscript is "in revision at *eLife*". Please let us know if you would like to pursue this option. (If your work is more suitable for medRxiv, you will need to post the preprint yourself, as the mechanisms for us to do so are still in development.)

Summary

This is a very comprehensive well-executed and well-presented study; it is very important in the Australian context where indigenous inhabitants have a markedly reduced life span and an excess of multiple diseases. The microbiome is influenced by diet, environment and lifestyle and impacts on host metabolism, immunity and chronic disease development. Microbiome studies in indigenous populations with high chronic disease burden are few. The authors have comprehensively studied the microbiome in two Torres Strait island populations with distinctly different food access and lifestyles and also have sociodemographic, dietary, anthropometric and extensive biomarker data. There are significant inter-community differences in the microbiome. The study provides insight into interactions between the microbiome, high sugar sweetened beverage consumption and inflammation and also between the microbiome, such beverages and high blood pressure.

Interestingly the paper goes against the current belief that western diet has worse effects on metabolic status than traditional diet. The discussion on these findings is balanced (including the role of high nutritional value / varied diet).

The authors argue that the models that have been developed using data from exclusively western populations may not be accurate in remote communities where chronic disease burden is disproportionately high. In this context, the study makes use of unique cohorts and datasets (host non-genetic factors, lifestyle parameters, food habit etc.) to identify the causal role of gut microbiome in modulating inflammation and disease risk.

More proteobacteriae were found in stool of subjects from Mer. Moreover, correlation between blood pressure (and sugar sweetened beverages ) with lachnospiraceae were observed in Mer subjects.

The study also provides a statistics framework to apply in future studies to understand the contribution of the microbiome to population health outcomes.

1) Although the study is well phenotyped, the major limitation of the study is the small sample size (n=50 versus 50). Given the dynamic nature of microbiome and the impact of multiple factors on microbiome, it is extremely difficult to conclude causal relationships with specific microbial traits and disease. It is essential that some of the associations that the authors identify should be worked out more clearly to prove the causal effect.

2) In the same line, for SEM, generally 100 samples or more are required (of course this depends on the context). Was there any sample size estimation done, e.g., by using Monte-Carlo simulation?

3) Intriguingly, as mentioned above, the authors find no association between Western dietary habits and increased chronic disease risk in Waiben residents. Except differences in age, no other explanation was given. By comparing to the Australian adult population, the authors state that 76% of the study population had overweight, 58% were classified as obese, and 39% had type 2 diabetes. Could this be split into two communities (Mer and Waiben)? Is having a large proportion of chronically diseased individuals in the study a confounder in the analysis? With the small sample size it difficult to establish causal relationships.

4) In the same vein, the central obesity (as observed in the Mer) would be expected to increase inflammatory markers and blood pressure. This should be expanded and clarified in Results and Discussion.

5) As the microbiota composition exerts its effects on the body not only via inflammatory cytokines, but also certainly via (diet-derived) plasma metabolites, it would be of importance to include (targeted) plasma metabolites in this paper. In that way, the reader will better understand the different microbe-diet interactions that drive these divergent aberrant metabolic effects.

6) The reviewers recommend to add gut microbial pathway analyses ( e.g., metacyc and biocyc, as well as Kegg functional annotation and pathway analyses) next to the currently performed metaphlan taxonomic strain identification in the metagenomic data.

7) Please clarify the reason for the groupings of particular biomarkers together.

8) Given the interest in the adverse effects of sugar sweetened drink consumption in many populations the authors could expand on the broader insights and relevance of their findings beyond these two communities.

9) Some of the key findings should be included in the Abstract.

10) The Discussion should start with stating the key findings.

---

## [Author Response]

The study also provides a statistics framework to apply in future studies to understand the contribution of the microbiome to population health outcomes.1) Although the study is well phenotyped, the major limitation of the study is the small sample size (n=50 versus 50). Given the dynamic nature of microbiome and the impact of multiple factors on microbiome, it is extremely difficult to conclude causal relationships with specific microbial traits and disease. It is essential that some of the associations that the authors identify should be worked out more clearly to prove the causal effect.

We recognize this is a limitation and we took care to highlight it within the Discussion section. We feel that it is important to stress though, that while the study only including 100 individuals, this represented a substantial proportion of the communities involved. The total population of Mer, for example, is only 453, which includes a substantial proportion of children and individuals who work on other islands.

We agree that, given the non-interventional nature of the study design, causality cannot be established. In parallel, we are undertaking basic research in an effort to establish causal interactions, including investigation of phenotype recapitulation in germ-free mice transplanted with faecal microbiota or individual bacterial populations. In addition, we have a number of interventional studies in populations at high risk of chronic inflammatory/metabolic disease that we also believe will provide important insight. However, when undertaking health research in Torres Strait Island communities, particular cultural considerations must be taken. We feel that the inclusion of human and basic/animal model work within the same publication would be inappropriate.

Throughout the text we highlight that observations are associative (in keeping with the overwhelming majority of human microbiome studies) and have updated the Discussion section to specifically highlight the issue of causality.

2) In the same line, for SEM, generally 100 samples or more are required (of course this depends on the context). Was there any sample size estimation done, e.g., by using Monte-Carlo simulation?

This analysis represents an ancillary study of the broader Well Persons Health Check (WPHC), a broad community health promotion and screening programme initiated in 1997 and conducted by the Torres and Cape Hospital and Health Service (TCHHS). Hence, then number of samples that could be obtained was constrained by the parent study sample size. One hundred participants represents almost 5% of the combined Mer and Waiben adult populations from which the samples were drawn. Given the observational nature of the study, the sample size was not able to be increased. It is possible for limitations of sample size to result in type 2 statistical errors, where true underlying associations may not be detected. Despite this tendency towards false negatives, we were still able to observe considerable biologically meaningful associations, suggesting that the sample size was adequate for the stronger, and hence more likely clinically relevant, associations to be observed.

To reflect this, we have discussed the limitation of sample size in the text, which reads: “A limitation of our study was its reliance on a relatively small sample, leading to the increased likelihood of type two statistical errors. However, we were still able to observe statistically significant associations that had biological relevance. Furthermore, it is important to note that the 100 participants in this study represented a substantial proportion of these two small island communities.”

3) Intriguingly, as mentioned above, the authors find no association between Western dietary habits and increased chronic disease risk in Waiben residents. Except differences in age, no other explanation was given. By comparing to the Australian adult population, the authors state that 76% of the study population had overweight, 58% were classified as obese, and 39% had type 2 diabetes. Could this be split into two communities (Mer and Waiben)? Is having a large proportion of chronically diseased individuals in the study a confounder in the analysis? With the small sample size it difficult to establish causal relationships.

In author response table 1, we present a breakdown of BMI and diabetes for the two communities and have included these separate data in the revised manuscript. Neither the distribution of body weight categories, nor the rate of type 2 diabetes, was significantly different between the two islands. The high proportions of overweight/obesity and chronic metabolic disease in our study populations is, unfortunately, representative of the Torres Strait Islander communities more generally.

**Author response table 1. resptable1:** 

	Waiben	Mer	Unadjustedp value
BMI			
kg/m^2^	31.1 (25.8, 37.9)	32.1 (25.5, 36.5)	0.893
Underweight (<18.5)	1 (2%)	1 (2%)	>0.99
Normal (18.5-24.9)	11 (22%)	11 (22%)	>0.99
Overweight (25-29.9)	9 (18%)	9 (18%)	>0.99
Obese (≥30)	29 (58%)	29 (58%)	>0.99
Type 2 diabetes			
Yes	15 (30%)	24 (48%)	0.10

Our study was focused on understanding the relationships between risk exposures, the intestinal microbiome, and health outcomes. An important contention was that, given the complexity of factors that could contribute to disease patterns, understanding the drivers of conditions such as obesity and type 2 diabetes required community-targeted approaches. As noted, one might expect to see a relationship between Western diet and chronic disease risk. The younger age of Waiben residents likely contributed to this pattern.

Generalisations, albeit well-supported ones, drawn from wider populations, might not hold true for specific populations with their own combinations of exposures and demographics. The prevalence of chronic disease within the study populations was therefore not a confounder, but simply a reflection of community characteristics. It is important to also highlight that, in keeping with this model, we are careful not to draw conclusions about microbial mediators of disease patterns beyond these communities.

We agree that establishing causality is challenging, both because of the relatively small numbers of individuals, but also because of the cross-sectional, non-interventional, nature of the study. We have taken care to not to suggest causality in the relationships that we report, and have reinforced this point in the revised manuscript.

4) In the same vein, the central obesity (as observed in the Mer) would be expected to increase inflammatory markers and blood pressure. This should be expanded and clarified in Results and Discussion.

As set out above, the complexity of risk exposures, demographic, and co-morbidity variables means that expectations around relationships established more widely were often not represented in our study populations (again, in support of the application of a community specific approach to understanding patterns of disease). We have expanded our discussion of this issue in the revised Discussion section.

5) As the microbiota composition exerts its effects on the body not only via inflammatory cytokines, but also certainly via (diet-derived) plasma metabolites, it would be of importance to include (targeted) plasma metabolites in this paper. In that way, the reader will better understand the different microbe-diet interactions that drive these divergent aberrant metabolic effects.

This is a good point, and one that we very much agree with. Clearly, with a study of this nature, there are a very large number of variables that we could have chosen to assess. Unfortunately, plasma metabolites were not included. However, this is something that we are including in follow-up longitudinal analysis in which we are attempting to identify effective interventions.

It light of this comment, we now highlight the analysis of gut and plasma metabolites as a potential focus for future studies.

6) The reviewers recommend to add gut microbial pathway analyses ( e.g., metacyc and biocyc, as well as Kegg functional annotation and pathway analyses) next to the currently performed metaphlan taxonomic strain identification in the metagenomic data.

In response to this suggestion, we undertook functional pathway analysis at a whole bacterial community level, and stratified pathways that showed significant differences in their representation between the two islands according to the contributory bacterial species. This analysis provided additional insight, including in identifying island-specific differences in bacterial functionality. We include these findings in the revised manuscript (Figure 2 and Figure 2—figure supplement 1). We also highlight in the Discussion the limitations of this work and suggest potential avenues of investigation that could be pursued in future studies.

7) Please clarify the reason for the groupings of particular biomarkers together.

The grouping of biomarkers was based on exploratory factor analysis. We include this information in the revised manuscript.

8) Given the interest in the adverse effects of sugar sweetened drink consumption in many populations the authors could expand on the broader insights and relevance of their findings beyond these two communities.

While we are cautious in commenting on the observed relationships beyond the study populations (other than in highlighting the importance of community-specific approaches) we agree that the harmful impact of sugar-sweetened beverages is considerable and unambiguous. The importance of reducing their intake as part of a strategy to improve health outcomes has been identified as being of central importance in Indigenous communities beyond the Torres Strait Islands, and is the focus coordinated campaigns in First Nations peoples in Australia and New Zealand.

In our revision, we have expanded the discussion of sugar-sweetened beverage consumption, including the potential mediatory role of the intestinal microbiome in their impact, as identified in our study. We further link this to how we might better understand important determinants of poor health whose impact might be influenced by intestinal microbiology and its relationship to human physiology.

9) Some of the key findings should be included in the Abstract.

This has now been included.

10) The Discussion should start with stating the key findings.

This has been addressed in our revision.